# Evaluation of antibiotic prescribing for the treatment of male community-acquired urinary tract infections using reimbursement data

Adrien Biguenet[1,2]*, Céline Slekovec[1,2], Kévin Bouiller[1,3], Xavier Bertrand[1,2]

**1** UMR-CNRS 6249 Chrono-environnement, Université de Franche-Comté, Besançon, France, **2** Hygiène Hospitalière, CHU de Besançon, Besançon, France, **3** Maladies Infectieuses, CHU de Besançon, Besançon, France

* abiguenet@chu-besancon.fr

## Abstract

### Introduction

Urinary tract infection (UTI) in men, although less common than in women, present specific diagnostic and therapeutic challenges. This study aims to evaluate the prescribing practices of general practitioners (GPs) for male UTI in France, focusing on adherence to guidelines.

### Materials and methods

We used an anonymous reimbursement database of antibiotics prescribed 15 days around an urine culture between September 2019 and August 2022 in a French region. Antibiotic prescriptions for male UTI were analysed according to adherence to national guidelines. Cluster analysis was used to identify different GP prescribing profiles. Prescription duration was assessed according to the number of antibiotic boxes delivered in the community pharmacy.

### Results

We included 7,816 urine culture prescriptions from 940 GPs for 6,457 male patients. We estimated compliance with French recommendations to be 55.7% for empirical treatment and 68.1% for documented treatment. GPs were divided into three clusters with different adherence to recommendations of 22%, 44% and 77%. Treatment duration for fluoroquinolones and cotrimoxazole was heterogeneous between GPs, but mainly too short.

### Conclusions

Our results suggest that our method could identify GPs who do not prescribe in accordance with recommendations and enable health insurance systems to target educational interventions to improve antibiotic prescribing practices.

**Data availability statement:** All relevant data are within the paper and its Supporting Information files.

**Funding:** The author(s) received no specific funding for this work.

**Competing interests:** The authors have declared that no competing interests exist.

## Introduction

The rise of bacterial resistance to antibiotics is a major health problem, leading to increased morbidity, mortality and hospitalization costs [1–3]. In 2022, France ranks 5th among European countries in terms of community antibiotic consumption [4]. This outpatient setting is responsible for 90% of antibiotic prescriptions in France, including 70% from general practitioners (GPs) [5,6]. Urinary tract infections (UTIs) are the second cause of antibiotic prescribing by GPs after respiratory tract infections, with an estimated 30–50% of prescriptions considered inappropriate [7,8].

Male UTIs are of particular interest for antibiotic stewardship. In France, the diagnosis of urinary tract infection in men is based on a urine dipstick test, which is systematically followed by a urine culture. The antibiotic prescribed must have a good diffusion in the prostate tissue and the treatment duration must be adapted (14 or 21 days depending on the antibiotic used or the presence of a prostatic abscess) [9]. As there is no specific clinical examination or imaging to diagnose prostate involvement, French guidelines recommend treating all male UTI as prostatitis with a fluoroquinolone (FQ), cotrimoxazole (TMP/SMX) or an intravenous third-generation cephalosporin (3GC) [9]. *Escherichia coli* is the predominant bacterium in UTIs [10] with a local resistance in the Doubs region in outpatient male urine culture in 2022 of 25.0% for TMP/SMX, 14.1% for ciprofloxacin (CIP), 17.8% for ofloxacin (OFL), 14.4% for levofloxacin (LVX) and 4.2% for 3GC [11]. Data on antibiotic prescribing in outpatient settings are global and do not include prescription indications. In France, some studies have created fictitious clinical cases to evaluate GP practice for UTIs due to the difficulty in obtaining real-world data [12–14], while in other countries, some studies have been able to assess actual prescribing practices [15–20].

We propose here an evaluation of GP prescriptions for male UTI in a French region using French health insurance data. The aim of the study was to assess GPs' adherence to French recommendations and to identify areas for improvement in antibiotic prescribing practices, particularly regarding antibiotic choice and treatment duration, by characterizing groups of GPs with similar prescribing patterns.

## Materials and methods

### Data collection

The Doubs health insurance fund (a French region with 550,000 inhabitants) provided us, following our specific request, with an anonymous reimbursement database of antibiotics prescribed 15 days around a urine culture, and dispensed by a community pharmacy between September 2019 and August 2022. Available data were single patient's anonymous number, age at time of treatment, prescriber's anonymous number, prescriber's specialty, patient's gender, date of urine culture prescription, date of urine culture realization, date of antibiotic prescription, Anatomical Therapeutic Chemical (ATC) code, and number of packages delivered. Data were accessed for research purposes on 13/10/2022.

## Definitions

Antibiotics prescribed less than two days after the urine culture realization were categorized as empirical treatment, whereas antibiotics prescribed two or more days after the urine culture realization were categorized as documented treatment. An antibiotic treatment was considered reassessed if an empirical treatment and a documented treatment were prescribed for the same urine culture, regardless of whether the antibiotic was the same or not.

## Inclusion and exclusion criteria

The major drawback of compiled antibiotic data from French health insurance is the absence of clinical data and treatment indications. To circumvent this problem, we have therefore excluded women for whom clinical data and indication are more diverse for measuring compliance with the recommendations. We only included male patients over the age of 15 and established a number of rules to enhance the value of the available data. First, we hypothesized that in case of suspected diagnosis of UTI, the patient would quickly go to the laboratory for urine analysis and receive either empirical treatment or documented treatment as soon as the results were available. To limit the risk of evaluating antibiotics that are not prescribed for UTIs, we only keep data for urine cultures with strict criteria: 1) the date of the urine culture prescription should not differ by more than 1 day from the date of the urine culture. We hypothesized that urine cultures prescribed for UTI would be more likely to be performed during this period; 2) the date of antibiotic prescription should be between −1 and +5 days after the urine culture (we assumed that treatment prescribed more than 5 days after the urine culture realization was probably for another indication); 3) we only included urine cultures prescribed by general practitioners, as we suspected that urologists prescribed urine cultures more often for reasons other than UTI; 4) we excluded urine cultures for which more than one antibiotic was prescribed in empirical treatment or in documented treatment (we hypothesized that these patients were either difficult to treat or receiving antibiotics for other reasons and were not representative of the general situation).

## Prescription evaluation

Antibiotic prescribing by GPs was assessed according to the French national recommendations [9], as antibiotic susceptibility test results were not available.. Empirical treatment with CIP, LVX or a 3GC was considered compliant to the recommendations. Empirical treatment with other antibiotics was considered as non-compliant. For documented antibiotics, the prescription of FQ (CIP, LVX or OFL), 3GC or TMP/SMX was considered compliant. The other antibiotic treatments were considered non-compliant.

## Estimation of the duration of treatment

The duration of treatment was not clearly available. However, we obtained the number of packages of antibiotics dispensed by community pharmacies. In France, antibiotics are not dispensed individually but in packages; therefore, patients do not receive the exact number of tablets. Our aim was to assess whether the duration of treatment with FQ and TMP/SMX for urinary tract infections was in accordance with recommendations, i.e., 14 days or 21 days depending on the presence of a prostatic abscess or urological disorder. However, due to the lack of clinical data, we were unable to determine whether the 21-day prescriptions were compliant.. Reassessed antibiotic treatments were not included in this analysis because it is not possible to calculate the actual effective duration of treatment, e.g., in cases where an antibiotic prescription was changed due to resistance detected by susceptibility testing. We used the usual treatment doses to calculate the duration of treatment. A package of LVX, OFL, or TMP/SMX in standard dosage provides up to 5 days of treatment, while CIP packages typically cover 6 days of treatment. For our analysis, we considered the CIP treatment period to 5 days, aligning with the 5-day treatment duration for LVX, OFL, and TMP/SMX. One package of CIP does not suffice for a 7-day treatment, and two boxes of CIP can provide either 7 or 10 days of treatment, similar to two packages of LVX, OFL, or TMP/SMX. In both scenarios, three packages are necessary to meet the recommended treatment duration.

## Clustering of GPs

We aimed to analyze the antibiotic prescribing patterns of GPs who had prescribed 10 or more empirical antibiotic prescriptions for positive urine culture. We also aimed to analyze duration of treatment pattern of GPs who had prescribed at least seven empirical or documented prescriptions of FQ or TMP/SMX (approximately 70% of the empirical or documented antibiotic prescriptions were FQ or TMP/SMX). We used hierarchical agglomeration clustering with the Ward method (ward.D2) to regroup GPs according to their prescribing profiles, minimizing intra-cluster variance [21,22]. The optimal number of clusters was determined using the elbow method, which assesses the within-cluster sum of squares to identify a point of diminishing returns, and the silhouette method, which evaluates cluster cohesion and separation to ensure meaningful groupings.

## Statistical analysis

Statistical analysis was performed using R software v4.2.3. We used as the unit of analysis the urine culture realization. The bootstrap method was used to estimate the 95% confidence intervals for both the mean and the median due to the non-parametric distribution of some of our data [23]. These intervals were determined from the 2.5th and 97.5th percentiles of the bootstrapped samples, with 1000 replacements and a significance level (alpha) of 0.05. Chi-square test was conducted to assess the independence between categorical variables. Results were considered statistically significant for $p < 0.05$.

## Results

### Study population

The database contained 95,650 prescriptions of urine cultures, in association with dispensing an antibiotic, for 48,988 patients between September 2019 and August 2022. Most prescriptions were made by GPs (n = 74,733; 78.1%) followed by urologists (n = 10,642; 10.9%) and midwives. (n = 2,977; 3.1%). After application of our exclusion criteria, we considered for analysis 7,816 urine cultures (8.2%) prescribed by 940 GPs (28.1%) for 6,457 male patients (13.2%) (Fig 1). The median age of patients was 63.5 years, IC 95% [62.9–64.0]. During the four years, the majority of patients were treated for one (n = 5,505; 77.5%) or two UTIs (n = 1,066; 16.5%). Each GP treated a median of 4 male UTIs [3.5–5]. A total of 5,744 (73.5%) of the prescriptions were empiric only, 1,302 (16.7%) were documented only and 770 (9.8%) were reassessed.

### Empirical antibiotic prescription

We included 6,514 empirical antibiotic prescriptions (5,744 prescriptions of empirical treatment only and 770 empirical treatments followed by a reassessment) from which 3,629 prescriptions (55.7%) were compliant to the French recommendations. CIP and LVX were the most prescribed antibiotics with 3,280 prescriptions combined (50.4%), followed by 3GC with 349 prescriptions (5.4%) (Fig 2). Among the non-compliant prescriptions, the four most commonly prescribed antibiotics were OFL (n = 758; 11.6%), amoxicillin with clavulanic acid (AMC) (n = 552; 8.5%), TMP/SMX (n = 539; 8.3%) and cefixime (n = 400; 6.1%). The mean age of patients who received a fluoroquinolone (59.8 years) was not statistically different from those who did not (58.9 years) (p = 0.051). To identify prescriber profiles, we performed clustering by hierarchical agglomeration. We selected the 230 GPs who had prescribed antibiotics for 10 or more UTIs, corresponding to 4,498 (69.1%) antibiotic prescriptions. Three clusters were created according to the specificity of the GP prescriptions. Cluster 1 (n = 25 GPs), cluster 2 (n = 125 GPs) and cluster 3 (n = 80 GPs) had rates of adherence to French recommendations of 21.9%, 44.4% and 77.0% respectively (Fig 3A). Cluster 1 was characterised by a higher proportion of OFL prescriptions (56.1% [48.5–64.0]) than cluster 2 (8.7% [6.9–10.6] or cluster 3 (2.4% [1.7–3.3] (p < 0.001). Cluster 2 was characterised by 11.6% [7.9–15.8] prescription of cefixime, 11.2% [8.8–13.8] prescription of AMC, 11.4% [8.4–14.7] prescription of TMP/SMX and 7.5% [5.3–10.1] prescription of AMX. Cluster 3 was characterised by a higher prescription of CIP and/or LVX (72.0% [68.1–75.7]) than cluster 1 (18.1% [13.1–23.5] or cluster 2 (37.6% [33.8–41.7] (p < 0.001).

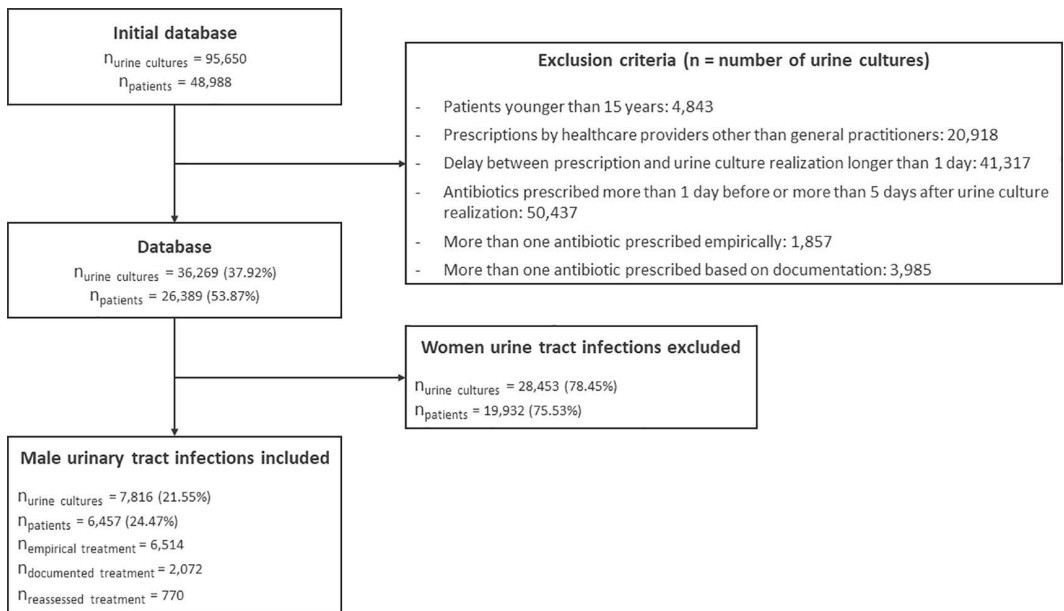

**Fig 1. Flowchart of exclusion and inclusion of male urinary tract infections from the Doubs French Health Insurance Database.**

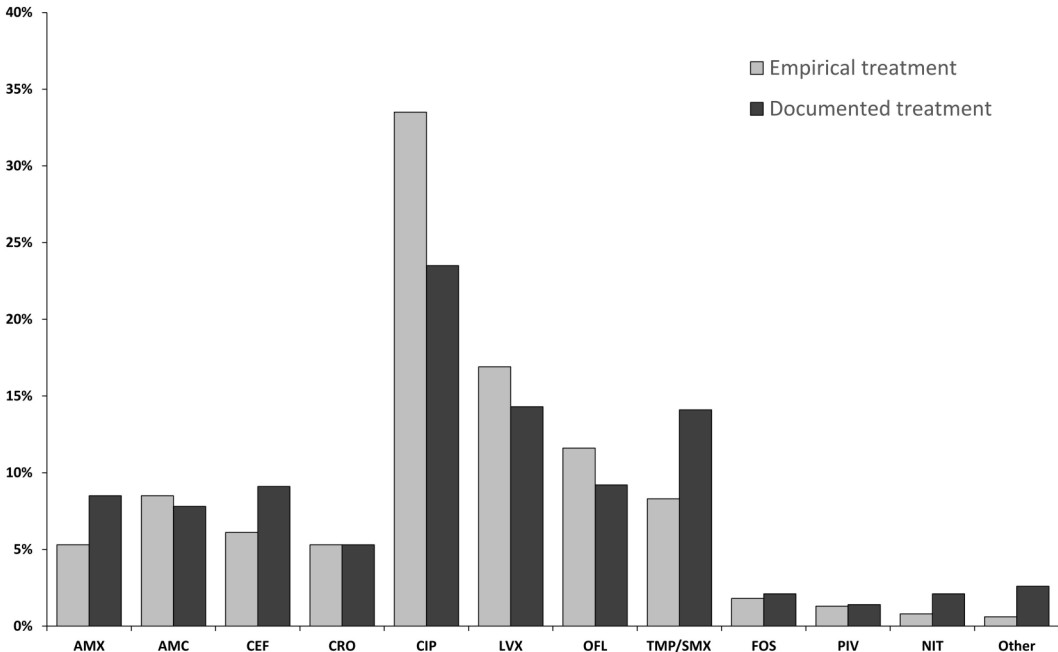

**Fig 2. Percentage of antibiotics prescribed empirically (n = 6,514) or documented (n = 2,072) by general practitioners for the treatment of urinary tract infections in men in the Doubs between 2019 and 2022.** AMX: amoxicillin; AMC: amoxicillin and clavulanic acid; CEF: cefixime; CRO: ceftriaxone; CIP: ciprofloxacin; LVX: levofloxacin; OFL: ofloxacin; TMP/SMX: cotrimoxazole; FOS: fosfomycin; PIV: pivmecillinam; NIT: nitrofurantoin.

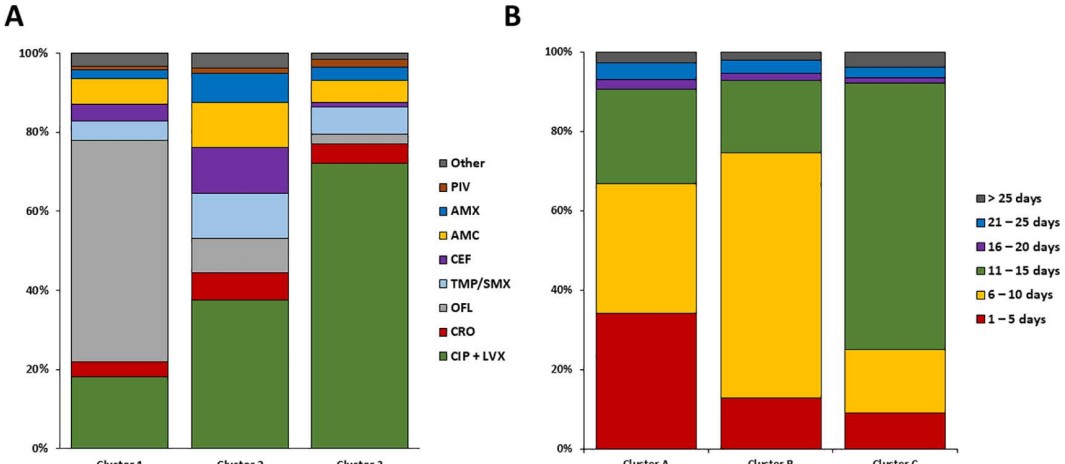

**Fig 3. Percentage of general practitioner (GP) prescriptions for (A) choice of antibiotic for empirical treatment within cluster 1 (25 GPs), cluster 2 (125 GPs) and cluster 3 (80 GPs), (B) duration in days of empiric-only and documented-only prescriptions for fluoroquinolones and cotrimoxazole within cluster A (103 GPs), cluster B (33 GPs) and cluster C (85 GPs).** The clusters were determined by hierarchical agglomeration clustering using the Ward method. AMX: amoxicillin; AMC: amoxicillin and clavulanic acid; CEF: cefixime; CRO: ceftriaxone; CIP: ciprofloxacin; LVX: levofloxacin; OFL: ofloxacin; TMP/SMX: cotrimoxazole; PIV: pivmecillinam.

## Documented antibiotic prescription

We included 2,072 documented antibiotic prescriptions (1,302 documented prescriptions and 770 reassessed prescriptions), of which 1,410 (68.1%) were considered compliant with French recommendations. Patients who received a documented prescription (64.0 years) were older than those who received empirical treatment (59.2 years) ($p < 0.001$). Compliance was higher for documented treatment compared with empirical treatment (68.1% vs 50.4%; $p < 0.001$). CIP remained the most prescribed antibiotic (23.5%). We observed some differences with empirical treatment with a decrease in the prescription of all FQ (CIP, LVX and OFL) from 62.0% to 47.0% ($p < 0.001$). In contrast, we observed an increase in the prescription of TMP/SMX (from 8.3% to 14.1%; $p < 0.001$), cefixime (from 6.1% to 9.1%; $p < 0.001$) and AMX (from 5.3% to 8.5%; $p < 0.001$).

## Treatment reassessment

We analyzed 770 empirical prescriptions with a reassessment from 379 GPs. Of these prescriptions, 454 (59.0%) involved a change of antibiotic (Fig 4) and 316 (41.0%) were the same as the empirical prescription. The median time between prescription of empiric treatment and documented treatment was 3 days. Cefixime and TMP/SMX prescriptions increased after reassessment (+4.4% and +7.9% respectively, $p < 0.001$), whereas prescriptions for FQ (−7.8%, $p < 0.01$) and ceftriaxone (−4.2%, $p = 0.012$) decreased.

## Duration of empirical treatment and/or documented treatment

We evaluated the duration of treatment with FQ (CIP, LVX, OFL) and TMP/SMX. We included 4,832 prescriptions, of which 4,072 (84.3%) antibiotics were empirical and 760 (15.7%) were documented. Treatment with a reassessment were not included in this analysis. CIP was the most frequently prescribed antibiotic (n = 2,256; 46.7%), followed by LVX (n = 1,142; 23.6%), OFL (n = 805; 16.7%) and TMP/SMX (n = 629; 13.0%). Prescription of antibiotics for 1–5 days was more frequent in the empirical group (n = 774; 19.0%) than in the documented group (n = 110; 14.5%) ($p < 0.001$). Conversely, prescriptions for 11–15 days were more frequent in the documented group (n = 360; 47.4%) than in the empirical group

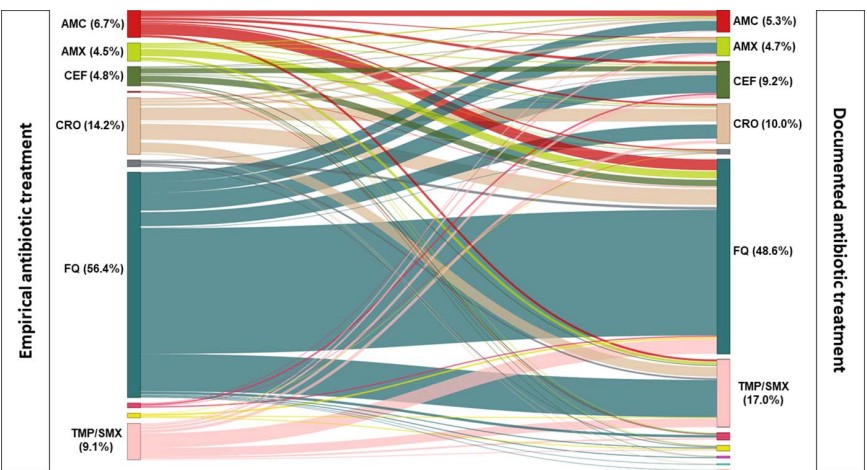

**Fig 4. Link between empirical prescribing and documented prescribing when the management of a male urinary tract infection benefited from a reassessment of antibiotic therapy (N = 770).** Left: empirical treatment. Right: documented treatment. The lines symbolize the link between empirical and documented treatment for a given case. The percentages (> 2%) indicate the proportion of each treatment prescribed as empiric or documented. AMX: amoxicillin; AMC: amoxicillin and clavulanic acid; CEF: cefixime; CRO: ceftriaxone; FQ: fluoroquinolones; TMP/SMX: cotrimoxazole.

(n = 1,641; 40.3%) (p < 0.001). There were no significant differences for the other treatment intervals. If we considered all empirical and documented treatments, LVX was prescribed more often for 11–15 days (n = 750; 65.7%) than other antibiotics (n = 1,251; 51.3%) (p < 0.001).

We included 221 GPs who had prescribed at least seven prescriptions of FQ or TMP/SMX, for a total of 3,339 antibiotic prescriptions (69.1%). GPs were classified into cluster A (n = 103 GPs), cluster B (n = 33 GPs) and cluster C (n = 85 GPs). The prescribing profile of the GPs varied according to the cluster (p < 0.001) (Fig 3B). Cluster A had 34.1% [27.1–40.8] of antibiotics prescribed for 1–5 days, 32.6% [29.2–36.2] for 6–10 days and 24.0% [20.2–28.2] for 11–15 days. Cluster B had 61.6% [55.7–67.8] prescribing for 6–10 days. Cluster C had 67.2% [62.8–71.4] of antibiotic prescriptions for 11–15 days. LVX was prescribed significantly more often in cluster C (32.6%) than in cluster A (16.5%) or cluster B (11.1%) (p < 0.001).

### Duration of empirical reassessed treatment

We aimed to compare the duration of treatment with FQ or TMP/SMX for GPs included in cluster A, B or C between empirical treatment only (n = 4,072) and empirical treatment with reassessment (n = 319) (Fig 5). The duration of reassessed prescriptions with no change in antibiotic (n = 136; 42.6%) did not differ from the duration of empiric treatment only (p = 0.077). However, the duration of reassessed prescriptions with a change of antibiotic (n = 186; 57.4%) was longer than the duration of reassessed prescriptions with the same antibiotic (p < 0.001).

### Discussion

Antimicrobial stewardship is a major public health issue. In France, the various campaigns to raise awareness of proper antibiotic use, aimed in particular at GPs, have had little impact on overall consumption. It therefore seems legitimate to try to identify GPs for whom information could improve their practice. On the basis of antibiotic and urine culture prescriptions for a clinical indication for which the recommendations are clear (i.e. UTIs in men), we attempted to determine, in a French region over a four-year period, the average level of adherence to the recommendations and to characterized the sub-populations of French GPs who complied poorly with the recommendations. We estimated compliance with French recommendations for antibiotic choice at 55.7% and 68.1% for empirical and documented treatment, respectively. We

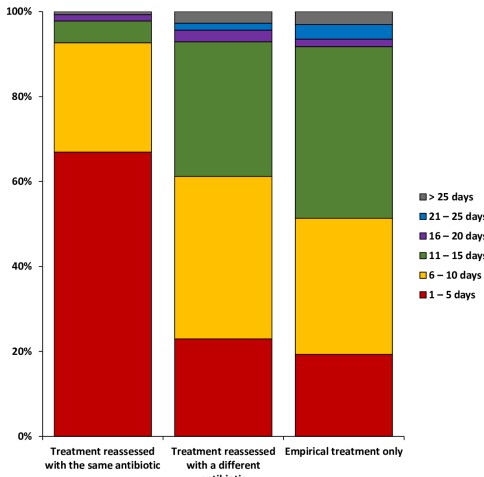

**Fig 5. Evaluation of the duration in days of fluoroquinolone or cotrimoxazole empirically prescribed by general practitioners selected from clusters A, B and C.** Treatment reassessed corresponds to empirical treatment with a documented prescription of the same or a different antibiotic.

observed heterogeneous prescribing patterns among GPs, with some prescribing in line with the French recommendations and others opting for non-recommended antibiotics such as cefixime or AMC. Treatment duration also varied between GPs, with one cluster prescribing 11–15 days of treatment, while others opted for 1–5 days or 6–10 days.

In the absence of specific treatment indications, we processed the data to ensure that the prescriptions analysed were exclusively for male UTI (or suspected infection), excluding other indications. Firstly, our demographic data closely matched those reported by Soudais et al [24]. They performed an analysis of the diagnosis and treatment of male UTI in another French region using data obtained from a different database. We obtained similar patient and GP characteristics with a median patient age of 63.5 years compared to 62.51 years in their study and 1.44 visits per GP per year for male UTIs, whereas we observed a median of 1 male UTI treated per GP per year in our study. Second, our results on adherence were consistent with those of other French real-world data studies [24–26]. Therefore, despite methodological differences, the proportions of each antibiotic class prescribed were very similar to our findings. Our high proportion of antibiotic prescriptions not in line with recommendations is probably due to significant variability in prescribing practices rather than a high rate of prescriptions for non-UTI infections.

The majority of prescriptions were too short, with 18.29% of prescriptions lasting 1–5 days and 32.31% lasting 6–10 days. One hypothesis was that GPs prescribed short treatment durations while waiting for urine culture results to adjust the antibiotic treatment if necessary. However, we observed a 4.5% difference between documented and empirical treatments for those lasting 1–5 days. In addition, no significant difference in treatment duration between empirical treatments alone and those reassessed with a change of antibiotic was observed. However, there is a significant difference in treatment duration when reassessed with the same antibiotic. These findings suggest that antibiotic treatments are generally too short, with only a small proportion, probably around 5% (as indicated by the difference between empiric and documented treatments), reflecting a real intention to reassess antibiotic treatment once urine culture results are available. Our results are consistent with the findings of Soudais et al, who found that 25% of antibiotic prescriptions lasted less than 7 days [24]. Interestingly, although 6–10 days of treatment is not recommended in France, a recent randomized clinical trial showed non-inferiority of 7 days of treatment compared with 14 days of treatment for non-febrile male UTI [27], but not for febrile male UTI [28].

We acknowledge that our study has several limitations. First, information on clinical indications was not available, although all male UTI are considered as prostatitis in France. Despite this limitation, our results are consistent with

previous studies. We hypothesized that the data filter we used allows us to account for antibiotic prescriptions in the context of suspected UTI. Second, we used antibiotic dispensing data to calculate prescription duration. It is possible that GPs prescribed the wrong dose of antibiotics, so a short prescription duration could be due to an incorrect dose. However, as a large proportion of treatments were significantly shorter than recommended, and as the standard dose is usually one or two tablets, this factor alone cannot explain the prevalence of treatments three times shorter (1–5 days) than recommended. Third, we did not have urine culture results to effectively assess adherence to documented treatment recommendations. However, Doubs is a region with a 4% prevalence of ESBL *E. coli*, which is the main bacterium involved in community UTI [11]. Therefore, a resistance to all FQ, TMP/SMX and 3GC together is rare and cannot explain why only 70% of prescriptions are in line with the French recommendations. Fourth, it is possible that some prescriptions were not included in the database if antibiotics were prescribed for male UTIs without a urine culture, even though a urine culture was recommended [24,29]. Comparing our results with data from other countries is challenging due to differences in national guidelines, particularly regarding antibiotic choice and treatment duration. The European Association of Urology (EAU) considers uncomplicated cystitis in men to be uncommon and distinguishes clearly between acute and chronic prostatitis—distinctions that are not made in the French guidelines [30]. Nonetheless, the EAU does recommend a 7-day treatment with trimethoprim-sulfamethoxazole in male cystitis. In contrast, guidelines in countries such as England [31], the Netherlands [32] or Denmark [33] recognize male cystitis as a clinical entity, allowing for the use of nitrofurantoin, trimethoprim, or pivmecillinam.

Interestingly, we were able to obtain similar data from previous studies of antibiotic prescribing decisions without having to survey GPs directly. Health insurance data provided us with comprehensive information from all GPs, not just those who agreed to participate, thus avoiding a major source of bias [34]. This approach allowed us to conduct analyses that yielded results consistent with previous studies using easily obtainable data that can be collected annually. Our approach could be used to more accurately assess prescribing for UTIs in men, rather than just the total number of antibiotics prescribed by GPs [35] or the prescribing of fosfomycin, nitrofurantoin and pivmecillinam as proposed [36,37]. Our findings suggest that further efforts are needed to improve antibiotic prescribing practices for male UTI, which is a relatively rare infection. In addition, GPs who did not prescribe according to recommendations for male UTIs could do the same for female UTIs. In the context of antibiotic stewardship programs, selective reporting of UTIs could influence antibiotic prescribing for documented treatments. However, solutions are needed to improve empirical prescribing and appropriate antibiotic duration. Identification of GPs who do not prescribe according to French recommendations could enable health insurance systems to target educational interventions and training to improve antibiotic prescribing practices.

## Supporting information

**S1 Table. Reimbursement data.**
(XLSX)

## Acknowledgments

We would like to thank the CPAM of Doubs for providing the databases on antibiotic consumption and urine culture prescription

## Author contributions

**Conceptualization:** Adrien Biguenet, Céline Slekovec, Xavier Bertrand.

**Formal analysis:** Adrien Biguenet.

**Methodology:** Kévin Bouiller, Xavier Bertrand.

**Supervision:** Céline Slekovec.

**Writing – original draft:** Adrien Biguenet.

**Writing – review & editing:** Céline Slekovec, Kévin Bouiller, Xavier Bertrand.

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
