## [Decision Letter · Decision Letter 0]

PONE-D-24-41708Evaluation of antibiotic prescribing for the treatment of male community-acquired urinary tract infections using reimbursement dataPLOS ONE

Dear Dr. Biguenet,

Thank you for submitting your manuscript to PLOS ONE. After careful consideration, we feel that it has merit but does not fully meet PLOS ONE’s publication criteria as it currently stands. Therefore, we invite you to submit a revised version of the manuscript that addresses the points raised during the review process.

We look forward to receiving your revised manuscript.

Kind regards,

Muhammad Qasim, Ph.D

Academic Editor

PLOS ONE

Journal Requirements:

2.  We note that there is identifying data in the Supporting Information file < S1 Table.xlsx>. Due to the inclusion of these potentially identifying data, we have removed this file from your file inventory. Prior to sharing human research participant data, authors should consult with an ethics committee to ensure data are shared in accordance with participant consent and all applicable local laws.

-Location data

Please remove or anonymize all personal information (<Patient ID>), ensure that the data shared are in accordance with participant consent, and re-upload a fully anonymized data set. Please note that spreadsheet columns with personal information must be removed and not hidden as all hidden columns will appear in the published file.

Reviewers' comments:

Reviewer's Responses to Questions

**Comments to the Author**

1. Is the manuscript technically sound, and do the data support the conclusions?

Reviewer #1: Yes

Reviewer #2: Yes

Reviewer #3: Partly

2. Has the statistical analysis been performed appropriately and rigorously? 

Reviewer #1: Yes

Reviewer #2: Yes

Reviewer #3: I Don't Know

3. Have the authors made all data underlying the findings in their manuscript fully available?

Reviewer #1: Yes

Reviewer #2: Yes

Reviewer #3: Yes

4. Is the manuscript presented in an intelligible fashion and written in standard English?

Reviewer #1: Yes

Reviewer #2: Yes

Reviewer #3: Yes

5. Review Comments to the Author

Reviewer #1: I have carefully read the manuscript entitled “Evaluation of antibiotic prescribing for the treatment of male community-acquired urinary tract infections using reimbursement data”. Despite the constraints associated with access to information (e.g. precise indication, duration of treatment, etc.), I note the contribution of this study to current community practice.

You will find below some comments / suggestion.

INTRODUCTION

- Line 33: You mention a duration of antibiotic treatment ranging from 14 to 21 days for male UTIs. However, as indicated in reference 9, this kind of infection, also known as prostatitis, require a treatment period of just 14 days: “A 14-day treatment duration is recommended for infections treated with fluoroquinolones or co-trimoxazole (SMX-TMP), except for the uncommon cases of abscess that may require a prolonged treatment (IV-C)”.

MATERIALS AND METHODS

- Line 49: Can you specify the origin of the data (e.g. extraction from the SNDS, specific request to the CPAM, other)?

- Line 51: Can you be more precise about the dates? Also, there is a discrepancy between line 51 and line 130 (August or September?).

- Line 54: Did you use the date on which the medicines were dispensed, in addition to the prescription date? This data could have been relevant for assessing dispensing conditional on the result of the urinalysis (e.g. two dispensations from the same prescription).

- Line 88: Have you taken into account patients' medication histories? In particular, the use of a quinolone in the last 6 months?

- Line 94: Can you specify whether the number of boxes dispensed per pharmacy relates to one visit per patient? For example, does a patient who gets one box on Monday and two on Thursday appear on 2 lines?

RESULTS

- Line 135: In relation to the previous comment, it would be interesting to check compliance with the recommendations for patients undergoing 2 UTIs in the near future.

- Line 140: specify that the data in the database is restricted to a given area (and not the whole of France).

- Line 167: I wonder about the presence of cefotaxime. Unless I am mistaken, this is a hospital reserve antibiotic, which should not be included in the reimbursement data for outpatient treatment.

- Figure 2: I'm not sure that a figure is more readable than a table. We only have percentages, so please include at least the sample size in the legend.

- Line 198: same remark for cefotaxime (Figure 4).

- Line 209: Why not assess the duration of ceftriaxone treatment? This is the only unit-dose antibiotic for which we know the exact duration of prescription.

DISCUSSION

- Line 297: There is a contextual element that may be partly responsible for your analysis, and that is the availability of antibiotics. In the post-COVID period, there have been tensions over various quinolones.

- Limitations: on reading the data table, don't you think that the short duration of treatment can be explained by monitoring prescriptions rather than patients? For example, if the same patient obtains a box of ciprofloxacin on 1 January and 2 others on 4 January, how can this be assessed on the basis of the available data?

- Openness: what solutions could you suggest to improve compliance with the recommendations?

The work carried out using data from the city reflects daily medical practice. As the authors and other studies have pointed out, there is still a great deal of progress to be made in reducing the risk of antibiotic resistance. Access to knowledge of the indication appears to be a limitation of the study, but very probably for dispensing these therapies in pharmacies.

Reviewer #2: Thank you for the opportunity to review this manuscript aiming to evaluate the prescribing practices of general practitioners (GPs) for male urinary tract infections (UTIs) in France, with a focus on adherence to guidelines. It is a well-written article that addresses an important topic. However, for a less familiar and more international audience, it would be helpful to clarify certain aspects. Some of the definitions used are not entirely clear to me, making it difficult to fully assess whether all methodological choices are justified. Providing more context and explanations for these definitions would enhance the accessibility and transparency of the study.

Line 53-54:

Urine culture realization: It is unclear whether this refers to the moment the urine sample arrived at the microbiology laboratory or when the culture result became available. If it refers to the former, a two-day cutoff for empirical treatment seems quite short.

Line 71:

What does the diagnostic process for UTIs in France look like? Is it primarily based on culture results, or are point-of-care tests (POCTs) like dipsticks also used? Adding more information on the diagnostic workflow in your introduction could provide better insight into the setting. This would also help clarify whether the chosen cut-offs—such as excluding urine cultures when antibiotics were prescribed more than one day before or more than five days after culture realization—are logical.

Do you know what percentage of UTIs are diagnosed with a urine culture? In other words, could selecting only cases with a urine culture introduce bias?

Line 74:

"The date of the urine culture prescription should not differ by more than one day from the date of the urine culture." Does the latter refer to the day the sample arrives at the laboratory?

Line 81:

This criterion might introduce bias by excluding prescriptions where treatment failure occurs. Wouldn’t treatment failure be more common for certain bacterial uropathogens/AB than others? Could you elaborate on this? Additionally, is it possible to distinguish between a switch due to treatment failure versus a switch based on urine culture results?

Line 90:

"For documented antibiotics, prescriptions of FQ (CIP, LVX, or OFL), 3GC, or SXT were considered compliant, while other antibiotic treatments were considered non-compliant." Is this classification based on the French guidelines? I find it difficult to understand why alternatives such as amoxicillin/clavulanic acid are not considered valid options in case of documented susceptibility.

Line 149 and beyond:

Providing exact numbers here would be helpful.

Discussion:

The French guidelines take an aggressive approach to treating UTIs in men, recommending long-term treatment and broad-spectrum antibiotics. However, there is little discussion on why general practitioners (GPs) do not adhere to these guidelines. Is there any evidence suggesting that GPs find the French guidelines too strict? Are they aware of studies like the one by Drekonja and similar research, and do they adjust their practice accordingly? Additionally, it might be valuable to analyze adherence rates across different patient age groups.

Reviewer #3: Thank you for the opportunity to review this interesting article draft. Male UTIs remain an understudied topic, making this manuscript particularly valuable to the scientific community. However, the manuscript currently lacks critical methodological details, particularly in the methods section, which hinders the reader’s ability to fully understand the data analysis process and limits the reproducibility of findings. That being said, I am confident that these issues can be addressed with revisions. I encourage the authors to refine their manuscript and look forward to reading an improved version soon.

Introduction:

• “30-50% considered inappropriate”—This is a strong claim. Could you clarify the specific context (region, country, disease setting, etc.)?

• Lines 40-41: The mention of fictional data seems unnecessary, as many studies use routinely collected health claims data for this purpose (examples are provided below). You might consider limiting this sentence to France or clearly specifying the research gap your study addresses.

• Lines 43-45: The stated aims do not seem to fully align with the analyses presented in the manuscript. You may want to expand on the specific objectives at the end of the introduction to help guide the reader through your analyses.

• Line 44: The phrase “identify GPs who might benefit…” could be misleading, as it implies that individual GPs will be singled out. A more appropriate phrasing could be: “identify areas for improvement in prescribing practices.” Similarly, in Lines 244-245, consider rewording: “It therefore seems legitimate to try to identify GPs for whom information could improve their practice.”

Methods and Results:

• Data Sources & Dataset Description:

1) Can you elaborate on the data sources? Did you access multiple datasets? If so, was data linkage performed? Please clarify the process.

2) The unit of analysis is unclear—was it the GP, the prescription, or the UTI episode? Please specify.

3) Consider listing all ATC codes used in your analysis, ideally in a table, e.g. in the supplementary table, and indicate whether each was classified as appropriate or inappropriate.

4) How were urine cultures analyzed? Did multiple labs contribute data? If so, did they follow the same analytical standards and positivity cut-offs? Please specify and give details.

5) Figure 1 presents the flow of included cultures, participants, and treatments, but it does not fully address the data source or unit-of-analysis issues. Consider clarifying this in the methods section.

Figure 1: What is meant by a “probabilistic antibiotic”? Please define this term clearly.

• Exclusion Criteria:

Exclusion criterion 4 (Lines 81-84) appears to contradict definitions provided in Lines 62-63. Could

you clarify this potential inconsistency?

• Unit of Analysis:

I don´t understand which was the unit of analysis. Some studies use UTI episodes as the unit of analysis—could this apply to your dataset?

• Antibiotic Reassessment:

1) Line 58: “An antibiotic treatment was considered reassessed if an empirical treatment and a documented treatment were prescribed for the same urine culture, regardless of whether the antibiotic was the same or not.”

2) Please specify the timeframe—how many days after the culture did you assume the antibiotic was prescribed in response to that culture result?

3) Line 88: Readers outside France may find these recommendations surprising. Consider expanding the introduction to provide a broader context on treatment guidelines in Europe and the US before focusing on France.

Relevant guidelines/recommendations:

Europe: EAU, German guidelines for uncomplicated UTI, NICE/SIGN

US: IDSA (though outdated), Kurotschka, P. K.; Gágyor, I.; Ebell, M. H. Acute Uncomplicated UTIs in Adults: Rapid Evidence Review. Am Fam Physician 2024, 109 (2), 167–174

• Assessment of Treatment Duration:

1) Line 98-99: “In the case of abscess or uncontrolled underlying urological disorder” - Is “absence” meant? What qualifies as “uncontrolled”? How was this information obtained, given that your dataset does not contain antibiotic indications? Did it contain comorbidities?

2) Line 99: “It is not possible to calculate the actual effective treatment duration.” Could you clarify why this is the case? If this is a limitation of your dataset, please explain; if it is a general issue, references would be helpful.

3) Lines 103-109: The explanation of treatment duration standardization is unclear. What method of standardization did you use (direct, indirect, other)? Please specify.

4) If one CIP package covers 6 days, how do two packages cover 7-10 days, if 2×6 = 12?

5) In the results, treatment duration is categorized as 1-5 days and 11-15 days—this categorization should be mentioned in the methods with justification.

• Clustering of GPs:

1) The selection criteria for GPs (≥10 or ≥7 prescriptions) need further justification and context.

2) The hierarchical agglomeration clustering method (Ward’s method) and the elbow/silhouette methodsrequire additional explanation regarding their purpose in this study.

• Statistical Analysis:

1) Mean and median of what variables and distribution? Also, usually SD and IQR or ranges are reported near to point estimates and need to be mentioned here. Please clarify.

2) Why was bootstrapping used instead of more standard methods (e.g., Wald CIs)?

3) Chi-squared test—what variables were tested for independence? Please provide a general explanation (e.g., antibiotic appropriateness y/n vs. X y/n?).

4) More details on clustering methods and relevant methodological references should be included in this section.

5) Specify the software version used for the analysis.

For reference, please review the level of methodological detail in the following studies:

https://journals.plos.org/plosone/article?id=10.1371/journal.pone.0312620

https://pubmed.ncbi.nlm.nih.gov/37487642/

https://pubmed.ncbi.nlm.nih.gov/35740228/

https://pubmed.ncbi.nlm.nih.gov/37974396/

https://pubmed.ncbi.nlm.nih.gov/33923682/

https://pubmed.ncbi.nlm.nih.gov/36823068/

https://pubmed.ncbi.nlm.nih.gov/32956399/

https://pubmed.ncbi.nlm.nih.gov/37528415/

These references may also be useful for strengthening your introduction and discussion.

Additional Points:

1) Line 35: Trimethoprim/sulfamethoxazole is commonly abbreviated as TMP/SMX—consider using this standard notation.

2) The terms “documented antibiotics” and “reassessed antibiotics” were difficult to follow; consider rewording or defining them more clearly.

3) Line 88: Avoid excessive use of non-standard abbreviations—this makes the text harder to follow. Consider using full drug names in each major section (introduction, methods, results, discussion).

4) Line 287: “…although there is only one context for male UTI in France.”—This sentence is unclear. Please clarify.

5) Line 289: Justify why the data filter used was appropriate, here you state opnly that you believe it was. You will need to convince the reader.

6) Line 290: You mention “dispensing data” - was this from a different database? Please specify the source (see comments above).

7) I would recommend copyediting/careful revision as I spotted some spelling/punctuation errors.

6. PLOS authors have the option to publish the peer review history of their article (what does this mean? ). If published, this will include your full peer review and any attached files.

**Do you want your identity to be public for this peer review?** For information about this choice, including consent withdrawal, please see our Privacy Policy .

Reviewer #1: **Yes: ** Arthur PIRAUX

Reviewer #2: **Yes: ** Tamara N. Platteel, MD PhD

Reviewer #3: No

---

## [Author Response · Author response to Decision Letter 1]

5 Apr 2025

Reviewer #1:

I have carefully read the manuscript entitled “Evaluation of antibiotic prescribing for the treatment of male community-acquired urinary tract infections using reimbursement data”. Despite the constraints associated with access to information (e.g. precise indication, duration of treatment, etc.), I note the contribution of this study to current community practice.

You will find below some comments / suggestion.

INTRODUCTION

- Line 33: You mention a duration of antibiotic treatment ranging from 14 to 21 days for male UTIs. However, as indicated in reference 9, this kind of infection, also known as prostatitis, require a treatment period of just 14 days: “A 14-day treatment duration is recommended for infections treated with fluoroquinolones or co-trimoxazole (SMX-TMP), except for the uncommon cases of abscess that may require a prolonged treatment (IV-C)”.

Response: We agreed. Line 33 was modified.

MATERIALS AND METHODS

- Line 49: Can you specify the origin of the data (e.g. extraction from the SNDS, specific request to the CPAM, other)?

Response: We added “specific request”

- Line 51: Can you be more precise about the dates? Also, there is a discrepancy between line 51 and line 130 (August or September?).

Response: We changed the date, line 130.

- Line 54: Did you use the date on which the medicines were dispensed, in addition to the prescription date? This data could have been relevant for assessing dispensing conditional on the result of the urinalysis (e.g. two dispensations from the same prescription).

Response: The CPAM did not provide us the date on which the medicines were dispensed.

- Line 88: Have you taken into account patients' medication histories? In particular, the use of a quinolone in the last 6 months?

Response: We did not take into account patients’ medical histories because the CPAM only provided us antibiotics associated with an urine culture. It is possible that patients were taking antibiotics for other indications that were not included in our data.

- Line 94: Can you specify whether the number of boxes dispensed per pharmacy relates to one visit per patient? For example, does a patient who gets one box on Monday and two on Thursday appear on 2 lines?

Response: We did not have this specific information. However, the number of boxes dispensed for a given patient was recorded separately based on the prescription date, the number of tablets per box, and the pharmaceutical company marketing the antibiotic

RESULTS

- Line 135: In relation to the previous comment, it would be interesting to check compliance with the recommendations for patients undergoing 2 UTIs in the near future.

Reponse: We agree that this would be interesting. However, the patients identified with two UTIs had infections spread over a period of four years. Therefore, it is likely that a significant proportion of these UTIs are not related and did not depend on compliance with the treatment of the first infection.

- Line 140: specify that the data in the database is restricted to a given area (and not the whole of France).

Response: Done as suggest.

- Line 167: I wonder about the presence of cefotaxime. Unless I am mistaken, this is a hospital reserve antibiotic, which should not be included in the reimbursement data for outpatient treatment.

Response: You are absolutely right. This was ceftriaxone prescriptions. We modified the manuscript.

- Figure 2: I'm not sure that a figure is more readable than a table. We only have percentages, so please include at least the sample size in the legend.

Response: The sample size has been added as suggested.

- Line 198: same remark for cefotaxime (Figure 4).

Response: Done as suggest.

- Line 209: Why not assess the duration of ceftriaxone treatment? This is the only unit-dose antibiotic for which we know the exact duration of prescription.

Response: We conducted the analysis for patients treated with ceftriaxone (without reassessment of treatment). For empirical treatments, we observed treatment durations of 1 day (30%), 6-7 days (28%), 10 days (8%), and 14 days (8%). For documented treatments, we observed treatment durations of 1 day (30%), 6-7 days (14%), 10 days (6%), and 14 days (28%). Without clinical information, we could not determine whether the 1-day treatment was prescribed for a urinary tract infection or a sexually transmitted infection. Therefore, we did not include these data in the manuscript.

DISCUSSION

- Line 297: There is a contextual element that may be partly responsible for your analysis, and that is the availability of antibiotics. In the post-COVID period, there have been tensions over various quinolones.

Response: Prescriptions of quinolones during the study did not vary.

- Limitations: on reading the data table, don't you think that the short duration of treatment can be explained by monitoring prescriptions rather than patients? For example, if the same patient obtains a box of ciprofloxacin on 1 January and 2 others on 4 January, how can this be assessed on the basis of the available data?

Response: Data were evaluated based on 'urine culture' rather than antibiotic prescriptions. In your example, if the patient had two prescriptions (one on January 1st and another on January 4th), the patient would be considered as having both a probabilistic treatment and a documented treatment. The treatment durations were combined. However, if the January 1st prescription was considered as documented, the January 4th prescription was not taken into account, as there is no reason not to prescribe the entire treatment in the initial prescription if it is already considered documented.

- Openness: what solutions could you suggest to improve compliance with the recommendations?

Response: We believe that targeted training focused on the actual prescribing practices that pose challenges, rather than a general reminder of the guidelines, would better engage physicians and improve compliance with the recommendations.

The work carried out using data from the city reflects daily medical practice. As the authors and other studies have pointed out, there is still a great deal of progress to be made in reducing the risk of antibiotic resistance. Access to knowledge of the indication appears to be a limitation of the study, but very probably for dispensing these therapies in pharmacies.

Reviewer #2:

Thank you for the opportunity to review this manuscript aiming to evaluate the prescribing practices of general practitioners (GPs) for male urinary tract infections (UTIs) in France, with a focus on adherence to guidelines. It is a well-written article that addresses an important topic. However, for a less familiar and more international audience, it would be helpful to clarify certain aspects. Some of the definitions used are not entirely clear to me, making it difficult to fully assess whether all methodological choices are justified. Providing more context and explanations for these definitions would enhance the accessibility and transparency of the study.

Line 53-54:

Urine culture realization: It is unclear whether this refers to the moment the urine sample arrived at the microbiology laboratory or when the culture result became available. If it refers to the former, a two-day cutoff for empirical treatment seems quite short.

Response: The urine culture refers to the moment the urine sample arrived at the microbiology laboratory. Typically, the antibiotic susceptibility test in private laboratories is available during the second day.

Line 71:

What does the diagnostic process for UTIs in France look like? Is it primarily based on culture results, or are point-of-care tests (POCTs) like dipsticks also used? Adding more information on the diagnostic workflow in your introduction could provide better insight into the setting. This would also help clarify whether the chosen cut-offs—such as excluding urine cultures when antibiotics were prescribed more than one day before or more than five days after culture realization—are logical.

Do you know what percentage of UTIs are diagnosed with a urine culture? In other words, could selecting only cases with a urine culture introduce bias?

Response: In France, the diagnosis of urinary tract infection in humans is based on a urine dipstick test (which has low sensitivity but good specificity), followed by a systematic urine culture. According to French guidelines, all patients should undergo a urine culture for diagnosis and antibiotic susceptibility testing. A sentence has been added to the introduction.

Line 74:

"The date of the urine culture prescription should not differ by more than one day from the date of the urine culture." Does the latter refer to the day the sample arrives at the laboratory?

Response: Yes

Line 81:

This criterion might introduce bias by excluding prescriptions where treatment failure occurs. Wouldn’t treatment failure be more common for certain bacterial uropathogens/AB than others? Could you elaborate on this? Additionally, is it possible to distinguish between a switch due to treatment failure versus a switch based on urine culture results?

Response: We do not have data on urine culture results. Therefore, it is not possible to distinguish between changes due to treatment failure and those due to antibiogram results. The aim of this study was to evaluate the prescribing practices of general practitioners. Treatment failures for urinary tract infections are relatively rare. Although our criteria excluded certain complicated cases, this is not a major issue, as the primary goal remains to provide an overview of prescribing practices.

Line 90:

"For documented antibiotics, prescriptions of FQ (CIP, LVX, or OFL), 3GC, or SXT were considered compliant, while other antibiotic treatments were considered non-compliant." Is this classification based on the French guidelines? I find it difficult to understand why alternatives such as amoxicillin/clavulanic acid are not considered valid options in case of documented susceptibility.

Response: Yes, this classification is based on the French recommendations (reference 9 in the article). Amoxicillin/clavulanic acid is not considered a good alternative due to the poor prostatic diffusion of clavulanic acid. In France, all urinary tract infections are considered to be prostatitis (line 37).

Line 149 and beyond:

Providing exact numbers here would be helpful.

Response: Done as suggested.

Discussion:

The French guidelines take an aggressive approach to treating UTIs in men, recommending long-term treatment and broad-spectrum antibiotics. However, there is little discussion on why general practitioners (GPs) do not adhere to these guidelines. Is there any evidence suggesting that GPs find the French guidelines too strict? Are they aware of studies like the one by Drekonja and similar research, and do they adjust their practice accordingly? Additionally, it might be valuable to analyze adherence rates across different patient age groups.

Response: We do not know exactly why GPs do not follow the current guidelines. It is often assumed that younger doctors are more likely to adhere to them, but we did not have access to their age in our study to verify this hypothesis. It also seems unlikely that GPs have the time to thoroughly read scientific studies, given the large number of guidelines they need to keep up with across all medical fields. However, this remains a supposition.

Regarding the influence of patient age, we compared fluoroquinolone use across different age groups. We initially hypothesized that, due to their adverse effects, GPs would be more reluctant to prescribe fluoroquinolones to older patients. However, we did not observe any statistically significant differences (line 151). We did not have other specific hypotheses regarding the impact of age on prescribing patterns.

Reviewer #3:

Thank you for the opportunity to review this interesting article draft. Male UTIs remain an understudied topic, making this manuscript particularly valuable to the scientific community. However, the manuscript currently lacks critical methodological details, particularly in the methods section, which hinders the reader’s ability to fully understand the data analysis process and limits the reproducibility of findings. That being said, I am confident that these issues can be addressed with revisions. I encourage the authors to refine their manuscript and look forward to reading an improved version soon.

Introduction:

• “30-50% considered inappropriate”—This is a strong claim. Could you clarify the specific context (region, country, disease setting, etc.)?

Response: References 7 and 8 refer to studies conducted in general practice in France and the USA. These numbers do not seem too high, considering that we identified 30% of inappropriate documented treatments and 45% for empirical treatments.

• Lines 40-41: The mention of fictional data seems unnecessary, as many studies use routinely collected health claims data for this purpose (examples are provided below). You might consider limiting this sentence to France or clearly specifying the research gap your study addresses.

Response: The manuscript has been modified to more clearly highlight the difficulties in obtaining real-world data in France.

• Lines 43-45: The stated aims do not seem to fully align with the analyses presented in the manuscript. You may want to expand on the specific objectives at the end of the introduction to help guide the reader through your analyses.

Response: The aims of the study had been modified.

• Line 44: The phrase “identify GPs who might benefit…” could be misleading, as it implies that individual GPs will be singled out. A more appropriate phrasing could be: “identify areas for improvement in prescribing practices.” Similarly, in Lines 244-245, consider rewording: “It therefore seems legitimate to try to identify GPs for whom information could improve their practice.”

Response: General practitioners can be identified individually by the French social security system in the Doubs region, allowing actions to be targeted at clusters of general practitioners who prescribe in a similar manner.

Methods and Results:

• Data Sources & Dataset Description:

1) Can you elaborate on the data sources? Did you access multiple datasets? If so, was data linkage performed? Please clarify the process.

Response: The CPAM provided us with two datasets (one every two years), which we simply merged. We believe it is not necessary to explain this in the manuscript.

2) The unit of analysis is unclear—was it the GP, the prescription, or the UTI episode? Please specify.

Response: Done as suggest line 125. The unit of analysis is the urine culture.

3) Consider listing all ATC codes used in your analysis, ideally in a table, e.g. in the supplementary table, and indicate whether each was classified as appropriate or inappropriate.

Response: Antibiotics considered compliant with French guidelines are listed in line 88 for both empirical and documented treatments. We do not find it necessary to list all other existing antibiotics simply to indicate that they are non-compliant, as this is already stated in the manuscript.

4) How were urine cultures analyzed? Did multiple labs contribute data? If so, did they follow the same analytical standards and positivity cut-offs? Please specify and give details.

Response: All laboratories in the Doubs region contributed to the data. In France, laboratories follow the same analytical standards and positivity cut-offs for urine culture analysis. We do not have information on which laboratory performed each analysis.

5) Figure 1 presents the flow of included cultures, participants, and treatments, but it does not fully address the data source or unit-of-analysis issues. Consider clarifying this in the methods section.

Response: We added in statistical analysis the unit of analysis (urine culture). Line 125.

Figure 1: What is meant by a “probabilistic antibiotic”? Please define this term clearly.

Response: “probabilistic antibiotic” has been replaced by “empirically”.

• Exclusion Criteria:

Exclusion criterion 4 (Lines 81-84) appears to contradict definitions provided in Lines 62-63. Could you clarify

---

## [Decision Letter · Decision Letter 1]

PONE-D-24-41708R1Evaluation of antibiotic prescribing for the treatment of male community-acquired urinary tract infections using reimbursement dataPLOS ONE

Dear Dr. Biguenet,

Thank you for submitting your manuscript to PLOS ONE. After careful consideration, we feel that it has merit but does not fully meet PLOS ONE’s publication criteria as it currently stands. Therefore, we invite you to submit a revised version of the manuscript that addresses the points raised during the review process especially by Reviewer 3.

We look forward to receiving your revised manuscript.

Kind regards,

Muhammad Qasim, Ph.D

Academic Editor

PLOS ONE

Reviewers' comments:

Reviewer's Responses to Questions

**Comments to the Author**

1. If the authors have adequately addressed your comments raised in a previous round of review and you feel that this manuscript is now acceptable for publication, you may indicate that here to bypass the “Comments to the Author” section, enter your conflict of interest statement in the “Confidential to Editor” section, and submit your "Accept" recommendation.

Reviewer #1: All comments have been addressed

Reviewer #3: (No Response)

2. Is the manuscript technically sound, and do the data support the conclusions?

Reviewer #1: Yes

Reviewer #3: Partly

3. Has the statistical analysis been performed appropriately and rigorously? 

Reviewer #1: Yes

Reviewer #3: Yes

4. Have the authors made all data underlying the findings in their manuscript fully available?

Reviewer #1: Yes

Reviewer #3: Yes

5. Is the manuscript presented in an intelligible fashion and written in standard English?

Reviewer #1: Yes

Reviewer #3: Yes

6. Review Comments to the Author

Reviewer #1: I sincerely thank the authors for taking into account all the comments made by the various reviewers. The manuscript is improved, and the results more robust.

I would like to emphasize once again the contribution of these results in relation to male urinary tract infections, which are much less studied than female urinary tract infections.

Reviewer #3: The manuscript improved with revisions, however I still find some points questionable.

At the editor´s discretion, some more changes might be useful. I think that although the French databases allow, in theory, to go back to the prescriber, researchers are usually not allowed to do that. Even if they are (e.g. because who is doing the study is the same group of people that decides on further training of individual physicians), a research study normally does not aim to go back to individual prescribers (or groups of them) for retraining. This aim is more consistent with a clinical audit. Therefore, again, I would suggest you to rephrase referring to an aim that is consistent with the results of your analyses, i.e. I would refer to the identification of areas of improvement or similar, which are, as pointed out in your summary of findings, antibiotic type and duration.

The authors pushed back some of my suggestions. This is fine to me, however, after reading the revised manuscript again, I still think that a more international contextualisation of male UTI treatment would be important, given the many observational studies on the topic published internationally. I agree that comparing guidelines was not an objective of the study. However, the discussion of a manuscript normally broadens the perspective comparing what was found to what is already known. As the explicit focus was on guideline adherence, a more international perspective on what in other contexts might be considered appropriate and why in other context GPs adhere or do not adhere to guideline recommendations would be useful and would make the manuscript more interesting for an international readership.

Lastly, I realised that the results of the urine culture were not available. What I understand here is that you were not provided with data on urine culture positivity (as S1 table suggests). What I don´t understand is how the authors assessed if the "documented" antibiotic was appropriate or not. This would need, in my opinion, further explanation in the methods section.

7. PLOS authors have the option to publish the peer review history of their article (what does this mean? ). If published, this will include your full peer review and any attached files.

**Do you want your identity to be public for this peer review?** For information about this choice, including consent withdrawal, please see our Privacy Policy .

Reviewer #1: **Yes: ** Arthur PIRAUX

Reviewer #3: No

---

## [Author Response · Author response to Decision Letter 2]

2 Jun 2025

PONE-D-24-41708R1

Evaluation of antibiotic prescribing for the treatment of male community-acquired urinary tract infections using reimbursement data

PLOS ONE

Reviewer #1: I sincerely thank the authors for taking into account all the comments made by the various reviewers. The manuscript is improved, and the results more robust. I would like to emphasize once again the contribution of these results in relation to male urinary tract infections, which are much less studied than female urinary tract infections.

Reviewer #3: The manuscript improved with revisions, however I still find some points questionable.

• At the editor´s discretion, some more changes might be useful. I think that although the French databases allow, in theory, to go back to the prescriber, researchers are usually not allowed to do that. Even if they are (e.g. because who is doing the study is the same group of people that decides on further training of individual physicians), a research study normally does not aim to go back to individual prescribers (or groups of them) for retraining. This aim is more consistent with a clinical audit. Therefore, again, I would suggest you to rephrase referring to an aim that is consistent with the results of your analyses, i.e. I would refer to the identification of areas of improvement or similar, which are, as pointed out in your summary of findings, antibiotic type and duration.

Response: We modified our aim as suggested by Reviewer 3 : “The aim of the study was to assess GPs’ adherence to French recommendations and to identify areas for improvement in antibiotic prescribing practices, particularly regarding antibiotic choice and treatment duration, by characterizing groups of GPs with similar prescribing patterns.”

• The authors pushed back some of my suggestions. This is fine to me, however, after reading the revised manuscript again, I still think that a more international contextualisation of male UTI treatment would be important, given the many observational studies on the topic published internationally. I agree that comparing guidelines was not an objective of the study. However, the discussion of a manuscript normally broadens the perspective comparing what was found to what is already known. As the explicit focus was on guideline adherence, a more international perspective on what in other contexts might be considered appropriate and why in other context GPs adhere or do not adhere to guideline recommendations would be useful and would make the manuscript more interesting for an international readership.

Response: We agree that a broader international perspective would enrich the discussion. However, our study did not investigate the underlying reasons why some GPs adhere to national recommendations while others do not. These reasons are likely multifactorial and context-dependent, and would require qualitative studies or targeted surveys, which were beyond the scope of our work.

We have added a section in the Discussion highlighting the differences in the diagnosis and management of male urinary tract infections across countries. These variations make it difficult to compare guideline adherence between healthcare systems.

• Lastly, I realised that the results of the urine culture were not available. What I understand here is that you were not provided with data on urine culture positivity (as S1 table suggests). What I don´t understand is how the authors assessed if the "documented" antibiotic was appropriate or not. This would need, in my opinion, further explanation in the methods section.

Response: We were not able to assess whether the prescribed treatment matched the antibiotic susceptibility test results, as urine culture data were not available. However, resistance rates in our region are low (with approximately 4% of ESBL-producing strains, which are often resistant to cotrimoxazole, third-generation cephalosporins, and fluoroquinolones). Based on these epidemiological data, we acknowledge that, in a small proportion of cases, physicians may have been unable to prescribe a treatment fully aligned with French recommendations. Nevertheless, this cannot fully explain why only 70% of prescriptions were in accordance with the guidelines. This point is addressed in the Discussion section, lines 304–309.

We explained in the Methods section that antibiotic treatment was evaluated based on the French national recommendations. We have now added that this evaluation was performed as antibiotic susceptibility test results were not available.

---

## [Editor Report · Decision Letter 2]

Evaluation of antibiotic prescribing for the treatment of male community-acquired urinary tract infections using reimbursement data

PONE-D-24-41708R2

Dear Dr. Biguenet,

We’re pleased to inform you that your manuscript has been judged scientifically suitable for publication and will be formally accepted for publication once it meets all outstanding technical requirements.

Kind regards,

Muhammad Qasim, Ph.D

Academic Editor

PLOS ONE
---

## [Editor Report · Acceptance letter]

PONE-D-24-41708R2

PLOS ONE

Dear Dr. Biguenet,

I'm pleased to inform you that your manuscript has been deemed suitable for publication in PLOS ONE. Congratulations! Your manuscript is now being handed over to our production team.

Kind regards,

on behalf of

Dr. Muhammad Qasim

Academic Editor

PLOS ONE